# In-Season Internal and External Workload Variations between Starters and Non-Starters—A Case Study of a Top Elite European Soccer Team

**DOI:** 10.3390/medicina57070645

**Published:** 2021-06-23

**Authors:** Rafael Oliveira, Luiz H. Palucci Vieira, Alexandre Martins, João Paulo Brito, Matilde Nalha, Bruno Mendes, Filipe Manuel Clemente

**Affiliations:** 1Sports Science School of Rio Maior–Polytechnic, Institute of Santarém, 2140-413 Rio Maior, Portugal; alexandremartins@esdrm.ipsantarem.pt (A.M.); jbrito@esdrm.ipsantarem.pt (J.P.B.); matildenalha@gmail.com (M.N.); 2Life Quality Research Centre, 2140-413 Rio Maior, Portugal; 3Research Center in Sport Sciences, Health Sciences and Human Development, Quinta de Prados, Edifício Ciências de Desporto, 5001-801 Vila Real, Portugal; 4Graduate Program in Movement Sciences, MOVI-LAB Human Movement Research Laboratory, Physical Education Department, School of Sciences, UNESP São Paulo State University, 17033-360 Bauru, Brazil; luiz.palucci@unesp.br; 5Falculty of Human Kinetics, University of Lisboa, 1649-002 Lisboa, Portugal; brunomendes94@hotmail.com; 6Escola Superior Desporto e Lazer, Instituto Politécnico de Viana do Castelo, Rua Escola Industrial e Comercial de Nun’Álvares, 4900-347 Viana do Castelo, Portugal; filipe.clemente5@gmail.com; 7Instituto de Telecomunicações, Delegação da Covilhã, 1049-001 Lisboa, Portugal

**Keywords:** acute/chronic workload ratio, high-speed running, in-season, non-starters, RPE, soccer, starters, training monotony, training strain

## Abstract

*Background and Objectives:* Interpretation of the load variations across a period seems important to control the weekly progression or variation of the load, or to identify in-micro- and mesocycle variations. Thus, the aims of this study were twofold: (a) to describe the in-season variations of training monotony, training strain and acute:chronic workload ratio (ACWR) through session ratings of perceived exertion (s-RPE), total distance and high-speed running (HSR); and (b) to compare those variations between starters and non-starters. *Materials and Methods:* Seventeen professional players from a European First League team participated in this study. They were divided in two groups: starters (*n* = 9) and non-starters (*n* = 8). The players were monitored daily over a 41-week period of competition where 52 matches occurred during the 2015–2016 in-season. Through the collection of s-RPE, total distance and HSR, training monotony, training strain and ACWR were calculated for each measure, respectively. Data were analyzed across ten mesocycles (M: 1 to 10). Repeated measures ANOVA was used with the Bonferroni post hoc test to compare M and player status. *Results:* The results revealed no differences between starters vs. non-starters (*p* > 0.05). M6 had a greater number of matches and displayed higher values for monotony (s-RPE, total distance and HSR), strain (only for total distance) and ACWR (s-RPE, TD and HSR). However, the variation patterns for all indexes displayed some differences. *Conclusions:* The values of both starters and non-starters showed small differences, thus suggesting that the adjustments of training workloads that had been applied over the season helped to reduce differences according to the player status. Even so, there were some variations over the season (microcycles and mesocycles) for the whole team. This study could be used as a reference for future coaches, staff and scientists.

## 1. Introduction

Monitoring of the training load in soccer has become popular, whereby two main dimensions of load are considered [1]: (i) internal and (ii) external. The external load can be considered as the physical demands that occur in the players in response to the implemented drill/task, while the internal load corresponds to the psychophysiological responses to the external load [2]. Different outcomes can be considered for each of the dimensions, although the rate of perceived exertion (RPE) and heart rate responses are the most used measures associated with internal load [3]. On the other hand, in soccer, the external load is typically characterized by the distance covered at different speed thresholds, or the inertial-derived measures such as accelerations/decelerations or composite variables (e.g., player load) [4].

Monitoring loads allows one to identify the consequence of training plans on the players and to individualize the analysis [5]. Although it is useful to look for accurate measures representing the impact in a training session [6], interpretation of the load variations across a period of time also seems to be important [7]. In fact, calculating workload measures is a part of the strategies to control the weekly progression or variation of the load, or to identify within-week variations [8]. Among the possibilities, acute load (representing the accumulated load during a week), chronic load (typically represented by the mean load in the past weeks), acute:chronic workload ratio (ACWR, representing the relationship between acute and chronic workloads) [9], training monotony (TM) (representing the variability of load within the week) and training strain (TS) (representing the variability of the load multiplied by the acute load) [10] are some examples of how to control load taking into consideration different measures.

Considering that some of these measures are sensitive to load fluctuations, it can be expected that participating or not participating in soccer matches may influence the workload measures reported for the players. For example, it is expectable that players with greater participation in matches present greater values of accumulated load and chronic load. However, as a consequence, players with less participation should be carefully managed to be prepared for participating in matches and coping with a spike in load. Despite the apparently obvious consequence of participating more or less in matches being related with different workload measures, reports on this matter are limited [11]. For example, similar comparisons between starters and non-starters regarding the workload measures of new body load and metabolic power were found [11]. In junior soccer players, it was also found that weekly internal and external load measures were also significantly greater in starters than in substitute players [12].

However, the above-mentioned results still need more research that provides some description about the workload measures’ variations in accordance with the level of participation of players in elite soccer. This should be further researched to provide information about how to manage players with match stimulus and to identify possible strategies to level the load with individualized training for those who are not playing. Based on that, the aims of this study were twofold: (a) to describe the in-season variations of TM, TS and ACWR through s-RPE, total distance and high-speed running (HSR); and (b) to compare those variations between starters and non-starters.

## 2. Materials and Methods

### 2.1. Subjects

Seventeen elite soccer players participated in this study. The players belong to a team that participated in the UEFA Champions League. They were divided into two groups: starters (*n* = 9, age 26.2 ± 3.5 years, 180.1 ± 6.8 cm and 78.7 ± 5.8 kg) and non-starters (*n* = 8, 24.5 ± 4.6 years, 182. ± 6.8 cm and 76.6 ± 4.3 kg). The inclusion criteria were regular participation in most of the training sessions (80% of weekly training sessions), while the exclusion criteria included lack of player information, illness and/or injury for two consecutive weeks. Goalkeepers were excluded from the study. The criteria to define starters and non-starters were assessed week by week against a player´s attendance time at the match and training sessions, and to be considered a starter, a player had to complete at least 60 minutes in three consecutive matches; players who did not achieve this duration were considered non-starters [13]. All participants were familiarized with the training protocols and signed informed consent prior to the investigation. This study was conducted according to the requirements of the Declaration of Helsinki and was approved by the Ethics Committee of Polytechnic Institute of Santarém (252020Desporto).

### 2.2. Design

Training load data were collected over a 41-week competition period, in which 52 matches occurred during the 2015–2016 in-season. The team used for data collection competed in four official competitions across the season, including the UEFA Champions League, the national league and two more national cups from their own country. For the purposes of the present study, all of the sessions carried out as the main team sessions were considered. This refers to training sessions in which both the starting and non-starting players trained together. Only data from training sessions were considered. Data from rehabilitation or additional training sessions of recuperation were excluded. This means that sessions after the match day were included whenever both starters and non-starters trained together, but other kinds of recovery training were excluded. This study did not influence or alter the training sessions in any way. Training data collection for this study was carried out at the soccer club’s outdoor training pitches. Total minutes of training sessions included the warm-up, main phase and slow-down phase plus stretching.

The season was organized into 10 mesocycles (M: 1–10). The number of training sessions, number of competitive matches and total training duration for starters and non-starters are presented in Table 1.

### 2.3. Internal Training Load Quantification

During training sessions, the CR10-point scale, adapted by Foster et al. was applied [14]. Specifically, thirty minutes after the end of each training session, players rated their RPE value using an app on a tablet. The scores provided by the players were then multiplied by the training duration to obtain the s-RPE [14,15]. The players were previously familiarized with the scale, and all answers were provided individually to avoid non-valid scores.

### 2.4. External Training Load Quantification

Global positioning system (GPS) units (Viper pod 2, STATSports, Belfast, UK) with 10 Hz frequency were used to monitor the training duration, total distance and HSR (above 19 km/h) for each player. For better satellite reception of the GPS antenna, the GPS unit was placed on the upper back between the left and right scapula through a custom-made vest. Previously, Beato et al. [16] positively tested the validity and reliability of linear, multidirectional and soccer-specific activities through this system. Thirty minutes before the start of a training session, all devices were turned on to acquire satellite signals and to provide synchronization between the GPS clock and the satellite’s atomic clock. After the training sessions, the Viper PSA software (STATSports, Belfast, UK) was used to download data and to clip the entire training session (i.e., from the beginning of the warm-up to the end of the last organized drill). In order to avoid inter-unit error, players wore the same GPS device in each training session.

### 2.5. Calculations of Training Indexes

Through s-RPE, total distance and HSR, the following variables were calculated: (i) TM (mean of training load during the seven days of the week divided by the standard deviation of the training load of the seven days) [11,17], (ii) TS (sum of the training loads for all training sessions during a week multiplied by training monotony) [11,17] and (iii) ACWR (dividing the acute workload, i.e., the 1-week rolling workload data, by the chronic workload, i.e., the rolling 4-week average workload data) [18,19,20,21,22].

### 2.6. Statistical Analysis

Data were analyzed using SPSS version 22.0 (SPSS Inc., Chicago, IL, USA) for Windows. Initially, descriptive statistics were used to describe and characterize the sample. The Shapiro–Wilk and Levene tests were used to test the assumption of normality and homoscedasticity, respectively. A repeated measures ANOVA was used with the Bonferroni post hoc test once variables obtained normal distribution (Shapiro–Wilk > 0.05), and the Friedman and Mann–Whitney tests were used for variables that did not obtain normal distribution in order to compare different M and groups. Hedge’s g effect size (95% confidence interval) was also calculated. Hopkins’ thresholds for effect size statistics were used, as follows: ≤0.2, trivial; >0.2, small; >0.6, moderate; >1.2, large; >2.0, very large; and >4.0, nearly perfect [21]. Results were considered significant with *p* ≤ 0.05.

## 3. Results

Figure 1, Figure 2 and Figure 3 show an overall view of the weekly average for TM, TS and ACWR calculated through the s-RPE, total distance and HSR across the in-season for starter and non-starter players. Overall, Figure 1 shows that the highest TM_s-RPE_ occurred in week 1 for both starters and non-starters (7.2 and 7.0 AU, respectively), while the lowest value occurred in week 19 for starters (1.5 AU) and week 2 for non-starters (1.5 AU). The highest TS_s-RPE_ occurred in week 41 for both starters (8498.0 AU) and non-starters (15,263.9 AU), while the lowest values occurred in week 30 for starters (110.2 AU) and week 19 for non-starters (1310.9 AU). The highest ACWR_s-RPE_ occurred in week 21 for starters (1.6 AU) and week 10 for non-starters (1.5 AU), while the lowest ACWR_s-RPE_ occurred in week 36 for starters (0.5 AU) and week 17 for non-starters (0.7).

Figure 2 shows that the highest TM_TD_ occurred in week 21 for both starters and non-starters (38.2 and 17.1 AU, respectively), while the lowest values occurred in week 2 for both starters and non-starters (2.0 and 1.9 AU, respectively). The highest TS_TD_ occurred in week 21 for starters (558,935.0 AU) and week 15 for non-starters (282,938.6 AU), while the lowest values occurred in week 36 for starters (35,441 AU) and non-starters (42,676.8 AU). The highest ACWR_TD_ occurred in week 10 for both starters (1.6 AU) and non-starters (1.6 AU), and the lowest ACWR_TD_ occurred in week 36 for both starters (0.7 AU) and non-starters (0.8 AU).

Figure 3 shows that the highest TM_HSR_ occurred in week 21 for starters (2.9 AU) and week 36 for non-starters (2.9 AU), while the lowest values occurred in week 20 for starters (0.7 AU) and week 39 for non-starters (0.8 AU). The highest TS_HSR_ occurred in week 4 for starters (3855.6 AU) and week 10 for non-starters (3578.0 AU), while the lowest values occurred in week 18 for starters (218.1 AU) and week 14 for non-starters (365.8 AU). The highest ACWR_HSR_ occurred in week 10 for both starters (1.6 AU) and non-starters (1.6 AU), while the lowest ACWR_HSR_ values occurred in week 9 for starters (0.4 AU) and week 4 for non-starters (0.4 AU).

Table 2 presents the average values and differences between starters and non-starters during the 10 mesocycles for all variables analyzed. There are no differences between the groups.

Figure 4, Figure 5 and Figure 6 show the differences between mesocycles for TM, TS and ACWR calculated through the s-RPE, TD and HSR across the in-season for the whole team.

Overall, Figure 4A shows that the highest TM_s-RPE_ occurred in M6 and the lowest value occurred in M5. There only was one significant difference for TM_s-RPE_ in M4 > M5 (ES = 0.17). The highest TS_s-RPE_ occurred in M1 and the lowest value occurred in M5. There was a significant difference in M1 > M5 (ES = 1.50); M3 > M5 (ES = 1.57); M4 > M5 (ES = 1.42); M5 < M8 (ES = −0.62) and <M10 (ES = −0.97).

Figure 4B shows that the highest ACWR_s-RPE_ occurred in M6 while the lowest ACWR_s-RPE_ occurred in M5. There were significant differences in M1 > M5 (ES = 1.63) and <M6 (ES = 7.60); M3 > M5 (ES = 11.75) and <M6 (ES = −10.69); M4 < M6 (ES = −1.42); M5 < M6 (ES = −8.75), <M7 (ES = −9.35), <M8 (ES = −9.25), <M9 (ES = −8.33) and <M10 (ES = −7.17); M6 > M8 (ES = −7.25) and >M10 (ES = 5.85).

Overall, Figure 5A shows that the highest TM_TD_ occurred in M6 and the lowest value in M1. There were significant differences in M1 < M2 (ES = −7.80), <M3 (ES = −5.70), <M4 (ES = −6.18), <M5 (ES = −3.81), <M6 (ES = −1.55) and <M7 (ES = −8.03); M2 < M4 (ES = −6.42); M4 > M7 (ES = −4.89) and M9 (ES = −0.93). The highest TS_TD_ occurred in M6 and the lowest value occurred in M2. There were significant differences in M1 < M2 (ES = −6.52), <M2 (ES = −5.35), <M3 (ES = −5.03) and <M10 (ES = −4.33); M2 < M4 (ES = −4.73), >M5 (ES = −2.92), >M7 (ES = −1.69); M3 > M5 (ES = −2.63), >M7 (ES = −1.63), >M9 (ES = −3.00) and M4 > M5 (ES = −1.98), >M7 (ES = −1.51), >M9 (ES = −2.00). Additionally, M7 < M10 (ES = −3.52).

Figure 5B shows that the highest ACWR_TD_ value occurred in M6, while the lowest value occurred in M5. There were significant differences in M1 > M2 (ES = −12.21) and >M5 (ES = −17.02). M2 < M3 (ES = −12.18), <M4 (ES = −12.05), <M6 (ES = −10.95), <M7 (ES = −13.75) and <M8 (ES = −13.42). M3 > M5 (ES = −12.99). M4 > M5 (ES = −15.64). M5 < M6 (ES = −14.30), <M7 (ES = −16.41), <M8 (ES = −25.59), <M9 (ES = −23.62) and <M10 (ES = −13.89).

Overall, Figure 6A showed that the highest TM_HSR_ occurred in M6 and the lowest value in M1. There were significant differences in M1 < M3 (ES = −5.42), < M6 (ES = −5.47). M2 < M6 (ES = −4.95). The highest TS_HSR_ occurred in M1 and the lowest value occurred in M5. There were significant differences in M2 > M6 (ES = 1.55), > M5 (ES = 0.16), > M7 (ES = 0.15), > M8 (ES = 0.32) and > M9 (ES = 0.40). M3 > M5 (ES = 0.15) and > M8 (ES = 0.19). M5 < M10 (ES = −0.79). Additionally, M7 < M10 (ES = −1.56).

In Figure 6B, the highest ACWR_HSR_ value occurred in M6 while the lowest value occurred in M5. There were significant differences in M3 > M5 (ES = −5.05). M5 < M6 (ES = −4.93), < M8 (ES = −5.75), < M9 (ES = −5.78) and < M10 (ES = −5.21).

## 4. Discussion

The main purpose of the current study was to provide a description regarding training monotony (TM), strain (TS) and acute/chronic workload ratio (ACWR) based on perceived exertion, total distance (TD) and high-speed running (HSR) measures collected across in-season soccer. A secondary goal was to compare the time-related behavior of such metrics among starter and non-starter players. Our results in an elite European soccer team squad showed the following: (i) in the mesocycle with a greater number of matches disputed, higher values of various indices occurred, including either monotony (s-RPE, TD and HSR), strain (i.e., only TD in this specific case) or ACWR (s-RPE, TD and HSR); (ii) for all parameters considered, there were no significant differences between starters and non-starters; (iii) despite the similarities observed, players with a distinct status showed peak or lower values in distinct moments of the monitored period for some markers (e.g., lower TM_s-RPE_ in the start and middle of the season, respectively, for non-starters and starters); (iv) higher monotony of perceived exertion was reported in the beginning, while for strain, it happened at the end of the season, independent of player status. In the following paragraphs, we will discuss the role of possible increase in match congestion on the data presented here while also accounting for the absence of differences between starters and non-starters and the common pattern of time-related variations in training monitoring parameters.

According to our results, the most intense period of games notably induced increases in monotony and ACWR (all variables) and concerning TS_TD_. Eight soccer fixtures were played across one month, rendering an average of at least two per week during the mesocycle. Indeed, this can characterize a full, congested schedule as per previous definitions [22,23]. Despite the monotony of s-RPE being more than twofold above the suggested threshold of 2 AU [10] as in the case of starters, the total duration of training sessions decreased for such players, while the same was not valid for non-starters. This is possible given the requirements of the latter to be more involved in active training/matches during that moment of the season and given the likely need to rotate players. In fact, congested fixture periods are linked to the possibility of inducing greater TS [8], whilst they can impair physical match performance [23,24,25] and raise injury risk [22,26]. However, the values for ACWR were all below 1.3, independent of player status (see Table 2), which, in theory, may not represent exacerbated injury likelihood [27], despite the fact that such a question lacks consensus to date (see, for example, Impellizzeri et al. [28]). Based on these assumptions, it seems that adjustments promoted during training may help avoid worst scenarios relating to management of players’ workloads across the most congested period of matches in a season. However, particular attention should be paid to non-starters since they presented a high monotony and no reduction in total training time as compared to previous ones and aligned with a partly higher strain (i.e., TS_TD_) during the intense period of games.

One key finding of the present study was that when players were grouped according to their playing status as starters or non-starters, no significant differences were detected. This can suggest that contemporary soccer training methods require players to respond to stimuli delivered in a homogenous way, i.e., irrespective of whether generally starting the games or not. Importantly, one previous work verified opposing results, considering training TM and TS from accelerometer-derived variables, where starters showed greater values compared to their non-starter peers [14]. In contrast, non-starters may experience greater overall in-game physical exertion as compared to starters or players who participated in a whole match [29], and a similar condition was verified considering the most demanding passages of play [30]. Reports both have [31] and have not [32] confirmed the match–training-load associations in soccer. Of note, although there was no statistical significance here in the comparisons depending on player status, starters and non-starters reached maximal and minimal values for various markers at distinct moments. This finding can be related to distinct demands placed over each player across the season owing to situational-induced variations [33] and their prominent non-linear usage. Taken together, these assumptions could indicate that monitoring players on an individualized basis seems necessary, accounting for whether players generally start games on the pitch or the bench. Notwithstanding, traditional measures of workload such as TD and HSR may not be sensitive enough to detect possible status-related differences in monitoring strain, monotony and ACWR in training routines.

The findings from our investigation may assist in understanding the role of player status in various parameters used to control training in soccer as well potentially serving as a benchmark for future prescriptions and monitoring. Regardless of playing status and considering just the s-RPE, training monotony peaked at the beginning of the period (week 1), while the strain reached the largest values at the end (week 41). Such observations are different when compared to a six-week congestive period, which found lower values of monotony and strain in the first week, but the highest values in the last week for both variables [34]. The present results also disagree with the idea that a high degree of strain is often achieved when there is no competition (e.g., pre-season) [35]. High monotony early in the period may be indicative of either a poor ability of athletes to recognize the initial training loads or a true heavy stimulus applied, making it difficult to cope with as per the common fitness status of players at that moment. For example, the training session durations or strain levels were not the highest in the first week, whilst this does not hold true for TMs-RPE. Indeed, prior off-season training is recognized to impair physical capacity aspects [36] and it may have contributed to the prime TMs-RPE outputs.

Aside from the aforementioned potential derived implications (e.g., informing conditioning professionals on the effects of playing status and provision of reference values), a number of limitations of the present investigation should be highlighted. With the ever increasing energy requirements in soccer, the data gathered here may be outdated to some extent. The mostly descriptive nature of the work may limit its practical application. The generalizability of the results to other teams/countries, competitive standards and ages is also not warranted and requires replication studies. Complete description of training drills in further research can facilitate field implementation. Finally, co-variables such as match location, results and opponent quality should be considered in future studies as previously recommended [33].

## 5. Conclusions

To summarize, here, we observed across in-season soccer that spikes in training monotony, ACWR and strain for both internal and external load parameters (except regarding strain) may occur during match congestion intensification in elite soccer. Most importantly, apart from the extreme values being slightly discrepant (i.e., highest/lowest outcomes of the monitored markers varied according to playing status), starters and non-starters behaved equally across the period, thereby suggesting a lack of differences between them in the adjustments of training workloads during the period. Finally, the progression of the training cycle phases elicited distinct responses of monitoring indices, such as the monotony of perceived exertion, which reached peak values at the early season, and major strain was reported at the end-season stage. The results suggest that the training load and management of load were properly addressed, despite some play-time differences across the season. Moreover, the present study shows that it is possible to have a congested mesocycle with eight matches with higher workloads (M6). In addition, this is the first study to report data for the 10 mesocycles of the in-season period and could be considered a reference for future studies.

## Figures and Tables

**Figure 1 medicina-57-00645-f001:**
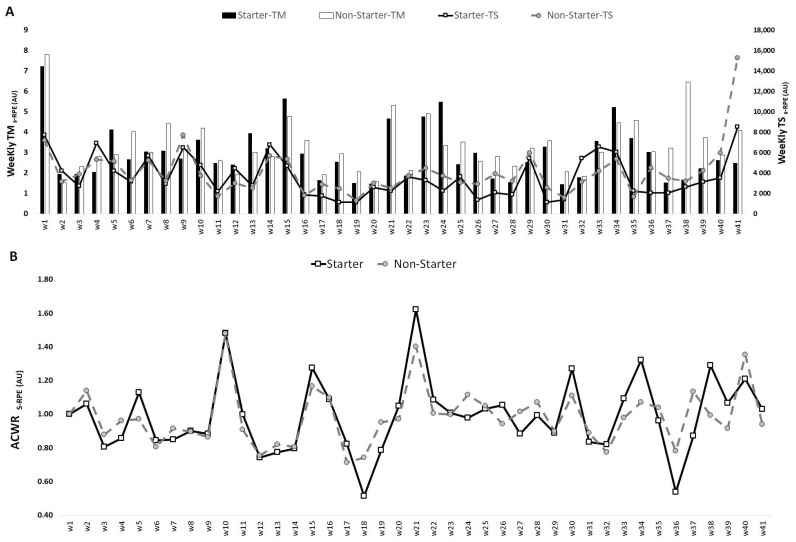
TM, TS (**A**) and ACWR (**B**) variations calculated through the s-RPE across 41 weeks for starters and non-starters.

**Figure 2 medicina-57-00645-f002:**
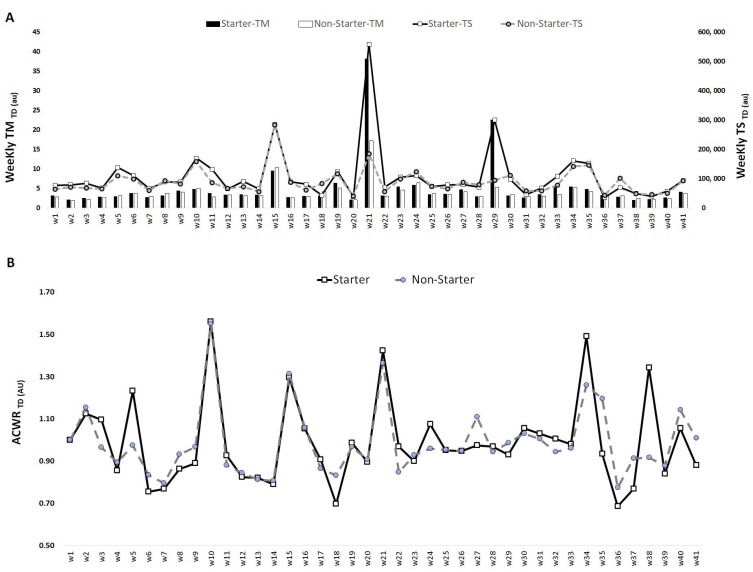
TM, TS (**A**) and ACWR (**B**) variations calculated through the total distance across 41 weeks for starters and non-starters.

**Figure 3 medicina-57-00645-f003:**
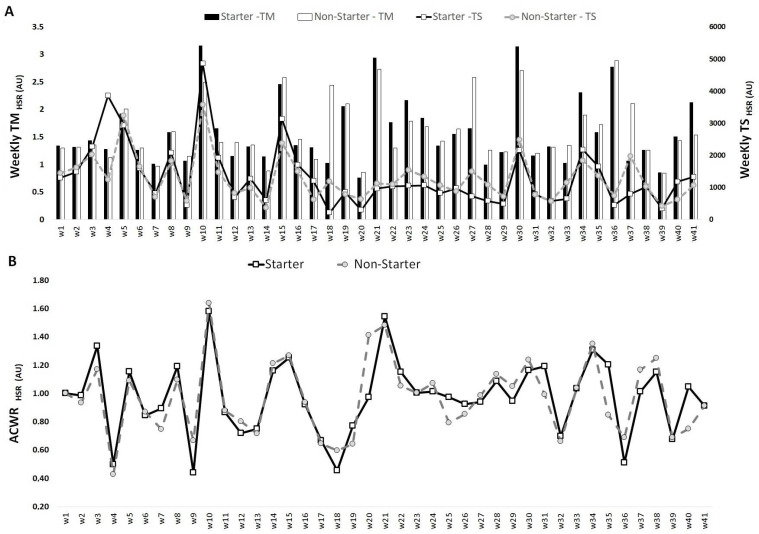
TM, TS (**A**) and ACWR (**B**) variations calculated through the HSR across 41 weeks for starters and non-starters.

**Figure 4 medicina-57-00645-f004:**
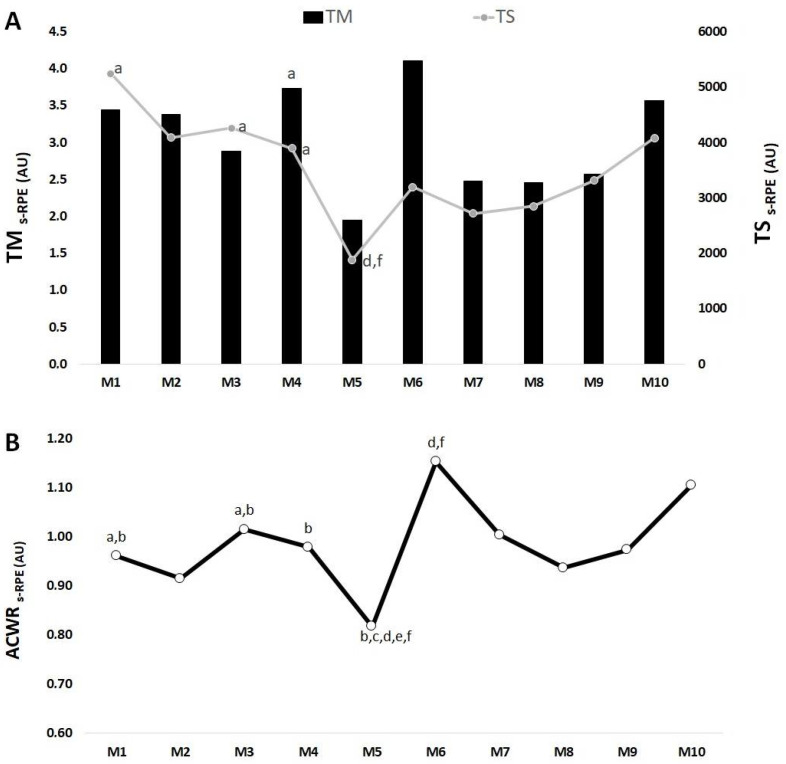
TM, TS (**A**) and ACWR (**B**) variations calculated through the s-RPE across 10 mesocycles for the whole team. M: mesocycle; a: difference from M5; b: difference from M6; c: difference from M7; d: difference from M8; e: difference from M9; f: difference from M10.

**Figure 5 medicina-57-00645-f005:**
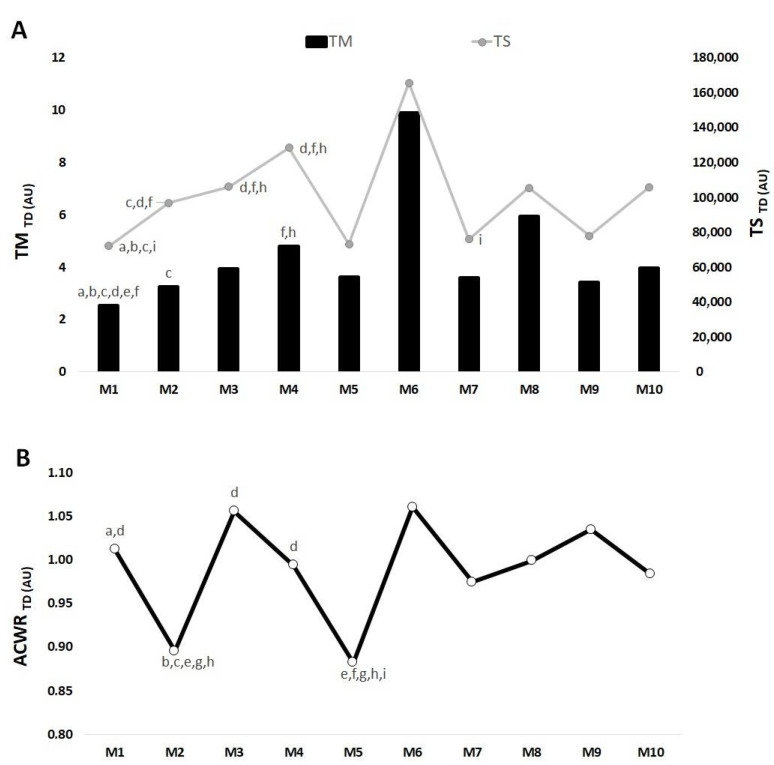
TM, TS (**A**) and ACWR (**B**) variations calculated through the total distance across 10 mesocycles for the whole team. M: mesocycle; a: difference from M2; b: difference from M3; c: difference from M4; d: difference from M5; e: difference from M6; f: difference from M7; g: difference from M8; h: difference from M9; i: difference from M10.

**Figure 6 medicina-57-00645-f006:**
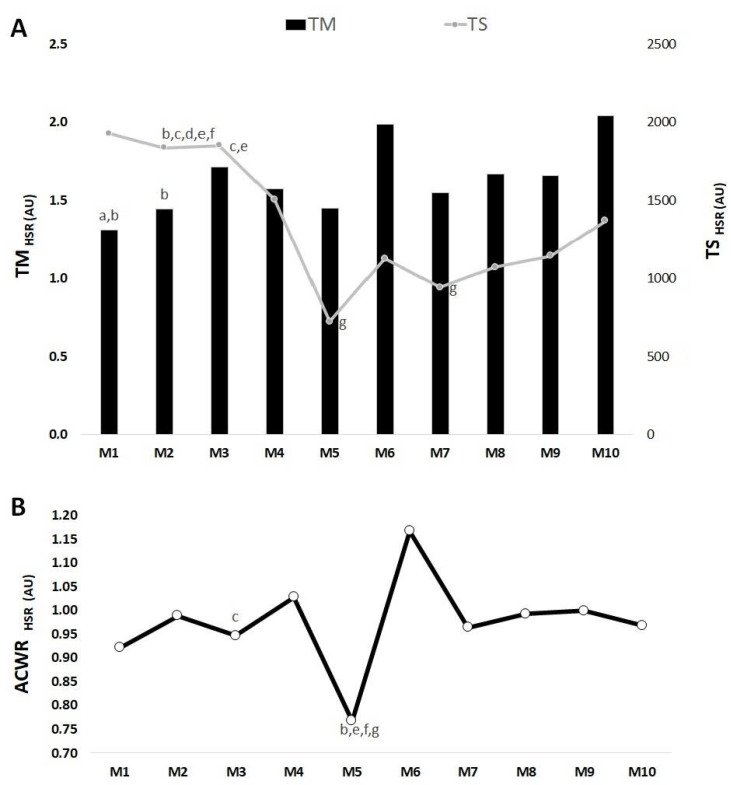
TM, TS (**A**) and ACWR (**B**) variations calculated through the HSR across 10 mesocycles for the whole team. M: mesocycle; a: difference from M3; b: difference from M6; c: difference from M5; d: difference from M7; e: difference from M8; f: difference from M9; g: difference from M10.

**Table 1 medicina-57-00645-t001:** Training sessions and number of competitive matches during the 41-week period.

Mesocycle (M)	M1	M2	M3	M4	M5	M6	M7	M8	M9	M10
Training sessions (*n*)	16	20	18	18	20	20	19	20	18	20
Session duration, total minutes, ST	1501	1778	986	1495	1062	864	1410	1519	1206	1227
Session duration, total minutes, NST	1585	1832	1029	1424	1197	1272	1599	1441	1358	1382
Number of matches (*n*)	4	5	4	5	6	8	5	4	7	4

ST = Starters; NST = Non-starters.

**Table 2 medicina-57-00645-t002:** Differences between starters and non-starters during the 10 mesocycles, mean ± SD.

Variables	M1	M2	M3	M4	M5	M6	M7	M8	M9	M10
TM s-RPE (AU), ST	3.3 ± 1.1	3.2 ± 0.8	2.8 ± 0.2	3.9 ± 0.4	1.8 ± 0.3	4.4 ± 1.0	2.2 ± 0.3	2.3 ± 0.3	2.5 ± 0.2	3.3 ± 0.7
TM s-RPE (AU), NST	3.6 ± 1.2	3.6 ± 0.8	3 ± 0.3	3.5 ± 0.4	2.1 ± 0.3	3.8 ± 1.1	2.8 ± 0.3	2.7 ± 0.4	2.6 ± 0.3	3.8 ± 0.7
TS s-RPE (AU), ST	5370.6 ± 881.1	3972.5 ± 900.1	4454.0 ± 510.3	4002.1 ± 445.4	1522.6 ± 486.3	2839.1 ± 505.2	2220.8 ± 367.0	2442.1 ± 538.0	3334.9 ± 667.1	4202.4 ± 949.1
TS s-RPE (AU), NST	5101.1 ± 934.5	4206.7 ± 954.7	4035.9 ± 541.2	3764.3 ± 472.4	2274.6 ± 512.6	3593.6 ± 535.9	3268.3 ± 389.3	8308.2 ± 570.6	3290.7 ± 707.5	3933.4 ± 1006.7
ACWR s-RPE(AU), ST	0.9 ± 0.02	0.9 ± 0.03	1.0 ± 0.01	1.0 ± 0.03	0.8 ± 0.03	1.2 ± 0.04	1.0 ± 0.03	1.0 ± 0.03	1.0 ± 0.04	0.9 ± 0.1
ACWR s-RPE (AU), NST	1.0 ± 0.03	0.9 ± 0.03	1.0 ± 0.01	1.0 ± 0.04	0.8 ± 0.03	1.1 ± 0.05	1.0 ± 0.04	0.9 ± 0.03	0.9 ± 0.04	1.0 ± 0.05
TM TD (AU), ST	2.7 ± 0.1	3.2 ± 0.1	4.1 ± 0.2	4.8 ± 0.3	3.7 ± 0.4	12.3 ± 2.7	3.7 ± 0.1	8.0 ± 2.5	3.6 ± 0.2	4.1 ± 0.3
TM TD (AU), NST	2.4 ± 0.2	3.3 ± 0.1	3.8 ± 0.3	4.8 ± 0.3	3.6 ± 0.4	7.2 ± 2.9	3.5 ± 0.1	3.7 ± 2.6	3.2 ± 0.3	3.8 ± 0.4
TS TD (AU), ST	76,836.5 ± 3760.5	100,533.8 ± 5541.1	113,493.5 ± 6692.5	132,192.8 ± 10097.1	71,403.2 ± 7200.7	199,545.0 ± 39571.2	75,732.0 ± 3461.4	127,443.4 ± 30,416.6	79,449.1 ± 5330.6	104,429.4 ± 9679.7
TS TD (AU), NST	66,845.5 ± 3988.6	92,677.9 ± 5877.2	97,736.2 ± 798.5	124,250.7 ± 10709.5	75,171.9 ± 7637.5	127,445.4 ± 41971.6	76,288.8 ± 3671.3	80,471.7 ± 32,261.7	76,347.3± 5653.9	107,630.7 ± 10,266.9
ACWR TD (AU), ST	1.0 ± 0.02	0.9 ± 0.03	1.1 ± 0.03	1.0 ± 0.02	0.9 ± 0.01	1.1 ± 0.03	1.0 ± 0.02	1.0 ± 0.01	1.0 ± 0.01	1.0 ± 0.03
ACWR TD (AU), NST	1.0 ± 0.02	0.9 ± 0.03	1.0 ± 0.03	1.0 ± 0.02	1.0 ± 0.01	1.1 ± 0.03	1.0 ± 0.02	1.0 ± 0.01	1.0 ± 0.01	1.0 0.03
TM HSR (AU), ST	1.3 ± 0.06	1.4 ± 0.1	1.8 ± 0.1	1.6 ± 0.1	1.3 ± 0.3	2.1 ± 0.1	1.4 ± 0.1	1.7 ± 0.1	1.7 ± 0.2	1.9 ± 0.3
TM HSR (AU), NST	1.3 ± 0.1	1.4 ± 0.1	1.6 ± 0.1	1.6 ± 0.1	1.6 ± 0.3	1.8 ± 0.1	1.7 ± 0.1	1.6 ± 0.2	1.6 ± 0.2	2.2 ± 1.4
TS HSR (AU), ST	2226.2 ± 482.7	1857.1 ± 288.3	2051.2 ± 308.4	1676.5 ± 280.9	641.0 ± 205.1	1008.5 ± 185.2	768.6 ± 145.6	1003.7 ± 179.8	1044.7 ± 212.0	1269.6 ± 165.9
TS HSR (AU), NST	1586.2 ± 512.0	1806.5 ± 305.8	1617.6 ± 327.2	1310.9 ± 297.9	811.9 ± 217.5	1253.6 ± 196.5	1132.8 ± 154.4	1140.1 ± 190.7	1245.5 ± 224.9	1479.0 ± 176.0
ACWR HSR (AU), ST	1.0 ± 0.04	1.0 ± 0.04	0.9 ± 0.03	1.0 ± 0.05	0.7 ± 0.06	1.2 ± 0.08	1.0 ± 0.05	1.0 ± 0.3	1.0 ± 0.3	1.1 ± 0.04
ACWR HSR (AU), NST	0.9 ± 0.04	1.0 ± 0.05	1.0 ± 0.04	1.0 ± 0.06	0.8 ± 0.06	1.2 ± 0.09	0.9 ± 0.05	1.0 ± 0.03	1.0 ± 0.03	1.0 ± 0.05

M = mesocycle; RPE = rating of perceived exertion; s-RPE = session rating of perceived exertion; TM = training monotony; TS = training strain; ACWR = acute:chronic workload ratio; AU = arbitrary units; ST = starters; NST = non-starters.

## Data Availability

The data presented in this study are available on request from the corresponding author.

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
