# Peer review of "In-Season Internal and External Workload Variations between Starters and Non-Starters—A Case Study of a Top Elite European Soccer Team"

_medicina, 2021, doi:10.3390/medicina57070645_

Round 1
Reviewer 1 Report
Dear Editor,
Firstly, thanks for allowing me to review this manuscript. We have carefully reviewed all sections of the paper (medicina-1228103). Please find enclosed my detailed suggestions to improve the quality of this research in the attached document.

Author Response
Reviewer 1
Dear Editor,
Firstly, thanks for allowing me to review this manuscript. We have carefully reviewed all
sections of the paper (medicina-1228103). Please find enclosed my detailed suggestions
to improve the quality of this research.
General comments
The purposes of this study were to (a) to describe the in-season variations of training
monotony, training strain, and acute: chronic workload ratio (ACWR) through session
rated perceived exertion (s-RPE), total distance and high-speed running (HSR); (b) to
compare those variations between starters and non-starters. The conclusion of the present
study indicates that he values of both starters and non-starters showed small differences
between them and thus suggesting that the adjustments of training workloads that had
been applied over the season helped to reduce differences according to the player status,
with variations over the season (microcycles and mesocycles) for the whole team.
Congratulations for a very good and valuable work with practical applications due to
future coaches, staff and scientist could be used it as reference. Instead, different minor
aspects should be addressed for considering the publication in Medicina MDPI journal.
Authors:
Dear reviewer,
We thank you for your valuable time and the helpful comments. We carefully addressed all of your concerns and suggestions in the following point-by-point statement. Amendments to the manuscript were made whenever necessary. All changes in the manuscript are marked in yellow.
Title
- The title of the study is specific but too long. Could the authors reduce it?. Reviewer
suggests to delete “training monotony, strain and acute/chronic workload ratio” and
use this title “In-season internal and external workload variations between starters and
non-starters. A case study of a top-elite European soccer team”. Include training
monotony, strain and acute/chronic workload ratio in keywords.
Authors: Dear reviewer, we thank you for the suggestion. We accepted it and changed accordingly.
- Authors should include “a case study” in the title. The data cannot be extrapolated to
other population.
Authors: As suggested, we changed accordingly.
Abstract
- Line 29: Previous to results include the statistical procedures.
Authors: Thank you for the suggestion. We add the following sentence: “Repeated measures ANOVA was used with Bonferroni post hoc to compare M and player status.”
- Line 33: Delete “In conclusion”.
Authors: Done.
- Line 38. Keywords: Include training monotony, strain and acute/chronic workload
ratio in keywords.
Authors: Dear reviewer, we already have those keywords. For better clarity we avoid the acronym, and use acute/chronic workload ratio.
Introduction
- The introduction and objectives are well-written and referenced. Congratulations for
the authors.
Authors: Thank you so much.
Methods
- Subjects: Please, authors need to detail if an informed consent was provided to the
participants and if the study follow ethical guidelines and was approved by a
bioethics institution. Include the sentence of the Institutional Review Board
Statement.
Authors: Thank you for your attention. We had that information in the institutional review board statement and informed consent statement and now added it in the end of subject section (2.1).
- Design: Authors need to clarify deeper if the complementary sessions after the match
day (MD+1 commonly) where starters and not-starters have different objectives
(recovery vs simulated match stimuli) were considered independently.
Authors: As we mentioned in the text, all the sessions carried out as the main team sessions were considered. This refers to training sessions in which both the starting and non-starting players trained together. Only data from training sessions were considered. Data from rehabilitation or additional training sessions of recuperation were excluded. This means that MD+1 sessions were included whenever both starters and non-starters trained together, but other kind of recovery training were excluded. We complete the section accordingly.
Results
- Authors need to improve figures 1, 2, 3, 4, 5 and 6. Numbers and titles are not
displayed correctly. Please, use special software to improve their quality.
Authors: We agree with you, and we changed the figures to improve quality. We also check all number, titles and legends. Thank you.
- Why did you use the rank of the reception of the players? This measure is
controversial. The use of the individual score of receptions is the only valid measure.
Authors: Dear reviewer, the metric related to the session rated perceived exertion was individually collected, through a tablet to avoid any kind of bias or non-valid scores as reported in section 2.3. We verified that a sentence was repeated, and we remove it accordingly.
- The table could be deleted, and the correlations included in a paragraph. A table for
this little information is not necessary.
Authors: Dear reviewer, we appreciate your suggestion, however we considered that the two tables presented have relevant information for better interpretation of the results. Table 1 describes training sessions and number of competitive matches during the 41-week period which allow the identification of congestive or non-congestive periods. Table 2 shows differences between starters and non-starters during the 10 mesocycles, mean ± SD. We considered important the presentation of the exact number, so they can be better interpreted by other researchers, coaches and practitioners. None of the tables present correlations.
- Why did not the authors think analyzing the differences of the individual score of
receptions between the different situations?
Authors: Dear reviewer, we thank you for the question that is quite pertinent. Indeed, we analysed the individual results, player by player. However, in order to analyse data for a scientific paper, we analysed all data according to player status (starters and non-starters).
Discussion
- The discussion section is well-organized. Congratulations
Authors: Thank you so much.
Conclusions and Practical Applications
- The conclusions are well-written. Instead, a specific practical applications section
with key message for coaches and sport scientists help to improve the quality of the
manuscript
Authors: Thank you. We add the following sentence to highlight strengths of our study: “The results suggest that training load and the management of load was properly ad-dressed, despite some play-time differences across the in-season. Moreover, the present study shows that it is possible to have a congested mesocycle with eight matches with higher workloads (M6). In addition, this is the first study to report data for the 10 mesocy-cles of the in-season period that could be considered a reference for future studies.”
Reviewer 2 Report
Despite some study limitations are clear and do not allow to generalize the results, we must congratulate the authors for the article developed, it is evident that the present article is an important contribution to researchers and trainers.
The research problem is clearly described, as well as its justification.
The methodology is clearly and objectively described.
The results are clearly presented, however I think authors should review the figures as they are not fully understandable.
Good discussion and study conclusions.
Author Response
Dear reviewers,
We thank you for your valuable time, kind words and comments. We improve the quality of the figures in the manuscript. We hope you think they are improved as well.
Best Regards,
The authors.